# Morpheme Analysis Associated with German Noun Plural Endings among Second Language (L2) Learners Using Event-Related Potentials (ERPs)

**DOI:** 10.3390/brainsci10110866

**Published:** 2020-11-17

**Authors:** Guiyoung Son

**Affiliations:** Department of Software, Sejong University, 209, Neung-dong-ro, Gwangjin-gu, Seoul 05006, Korea; sgy1017@sejong.ac.kr; Tel.: +82-2-3408-3847

**Keywords:** event-related potentials (ERPs), second language learning, electroencephalogram (EEG), left anterior negativity (LAN), P600, N400, German plural endings, Dual Mechanism Model

## Abstract

This paper aims to examine the morpho-syntactic process of noun plural endings, “-n” and “-s”, in adult second language (L2) learners using event-related potentials (ERPs). German noun plural endings consist of many inflectional forms. They are one of the difficulties faced by German L2 learners. We recorded an electroencephalogram (EEG) study of German L2 learners by dividing study subjects into low and high L2 learners according to the learning level. We examined what ERP components were associated with L2 language processing. All participants were Korean German L2 learners who had achieved varying levels of proficiency. As a result of our analysis, we confirmed different morpho-syntactic processing between the two groups. First, N400 was detected at any learning level. It confirmed language processing supportive of the Full-Listing Model for irregular endings. Second, we confirmed left anterior negativity (LAN), as detected in both low and high proficiency L2 learners. LAN is supportive of a Full-Parsing Model for regular endings, as it was detected in both low and high proficiency L2 learners. However, P600 was detected in highly proficient L2 learners only. It implies that high proficiency learners differ from low proficiency L2 learners. P600 is processed in a reparsing process after recognition of grammatical errors. Based on this result, more active use of a Dual Mechanism Model is possible as learning levels improve. It confirms that improvement in L2 learners results in an approach to cognitive processing similar to that of German first language (L1) speakers.

## 1. Introduction

Humans naturally learn a first language (L1) during development unless we have a specific congenital disorder or are restricted by environmental conditions. However, learning a second language (L2) is usually done after the critical period, except in bilingual individuals. Adult L2 learners have trouble with grammar and pronunciation when we learning new words after a certain critical period [1,2]. In particular, adult German L2 learners have difficulty using inflectional morphemes in verbs, nouns (pronouns), and adjectives. It is caused by grammatical systems that are different from their L1 language.

German plural noun endings can be classified into at least five types, including a zero morpheme, ‘-ø’, ‘-e’, ‘-er’, ‘-(e)n’, and ’-s’. Some researchers have asserted that there are eight types, including umlauts such as ‘-ø’, ‘-e’, and ‘-er’ [3,4,5,6]. This specificity causes some of the difficulties faced by adult German L2 learners. For example, Marcus et al. [7] showed that the ‘-s’ plural falls into the default cluster and applies when the phonological environment does not permit any other plural allomorph. Clahsen [6] showed that plural noun endings except ‘-s’ plural nouns create plural noun forms based on a mental lexicon, while the “-s” plural is created in new plural forms and is not stored in our mental lexicons. 

There are three types of morpho-syntactic processing models in second language processing: a Full-Listing Model, a Full-Parsing Model, and a Dual Mechanism Model. The models depend on structural analysis of the extent of lexicon familiarity necessary to accomplish morpho-syntactic processing. 

First, the Full-Listing Model states that a whole word is stored as the combined form of a stem and an affix. Thus, whole words are stored in the mental lexicon with no morpho-syntactic structural analysis [4,8,9,10]. In other words, all inflectional words are represented as independent and specific information in the mental lexicon and produced if necessary.

Second, the Full-Parsing Model states that whole words are represented and produced by structural analysis during morpho-syntactic processing [11,12]. Whole words include structural analysis with a stem and affix classification as the first language processing stage. For example, the singular nouns, “Kind” or “Ei” have the morpheme “-er”, indicating the singular form of the word, and the plural ending saved separately. Therefore, we can produce “Kinder” or “Eier”, which are represented via combining “Kind” + “-er” = “Kinder”, or “Ei” + “-er” = “Eier”.

Lastly, The Dual Mechanism Model combines the Full-Parsing Model and the Full-Listing Model. This model has a dual structure with two distinct representational mechanisms: the Full-Listing Model has some word-based operations stored in the mental lexicon with no structural analysis, and the Full-Parsing Model is a set of rule-based operations that are represented and produced from words in the mental lexicon [13,14,15]. In other words, Penke and Krause [16] presented that irregular forms are stored as fully inflected forms in the mental lexicon, whereas regular inflection is rule-based and is used by default whenever a stored irregular form cannot be retrieved from the mental lexicon. According to previous studies, inflectional forms are divided into regular and irregular forms. During morpho-syntactic processing, the regular form obeys the Full-Parsing Model, while the irregular form obeys the Full-Listing Model [17]. Many event-related potential studies, including one done by Münte et al. [18], have attempted to characterize these models. Recently, various languages, including English, have also been studied [6,15,18,19,20,21]. Despite the inflection systems of these languages being different from English, the results of these studies support the Dual Mechanism Model, based on the use of a different morpho-syntactic processing model in both regular and irregular forms.

Electroencephalography (EEG) is a neuroimaging method involving positron emission tomography (PET), functional magnetic resonance imaging (fMRI), and magnetoencephalography (MEG). Electroencephalography (EEG) is an important functional neuroimaging tool for studying the temporal dynamics of the human neural circuit. EEG has been used in many studies because it has relatively high accessibility. 

Event-related potentials (ERPs) are derived from EEG and measured by a brain response that is the direct result of a specific sensory, cognitive, or motor event. The results are usually presented as ERP components such as early left anterior negativity (ELAN), left anterior negativity (LAN), P300, N400, P600, etc. The components are detected when processing specific behavioral tasks. For example, we investigated the language-dependent component by averaging across many different words from several different ranges. ERPs are useful for studies of perceptual and cognitive processes related to language comprehension. In addition, since ERPs are valuable in generating information on the timing and brain activation in language processing, the technique is particularly useful in studying L2 processing rather than L1 processing [22,23].

ERPs related to language processing include early left anterior negativity (ELAN), left anterior negativity (LAN), N400, and P600. These component signals are characterized by polarity, latency, amplitude, brain region, functional description of a task, etc. The ELAN that occurs at a 150–250 ms latency is often lateralized over the left frontal lobe. The LAN also occurs as anterior negativity, but it occurs slightly later than the ELAN, at 300–500 ms. Both components are related to morpho-syntactic errors. N400 occurs at a latency of 300–500 ms and is often lateralized over the central and parietal lobes. It reaches a maximum of around 400 ms in a negative waveform. N400 occurs in the same time window as LAN but at different electrode sites. N400 relates to the integration of meaning [24,25]. Lastly, P600 occurs at 600–800 ms or lasts for several hundred milliseconds and is often lateralized over the centro-parietal lobes. P600 shows the greatest positivity peak at about 600 ms after stimulus onset. P600 is related to syntactic violation of syntactically complex structures [26,27].

The current study examines German plural noun endings in Korean German L2 learners. To accomplish the purpose of the study, we recorded ERPs arising from plural noun ending processing in adult German L2 learners and compared them to the results from previous studies in German L1 speakers. Here, we examine new findings on German inflection in adult German L2 learners using the violation paradigm. We specifically examine plural noun endings in German, which allow us to compare regular and irregular inflections. 

If morpho-syntactic processing in adult German L2 learners appears similar to language processing in German L1 speakers, regular or irregular forms of plural noun endings are processed based on the Dual Mechanism Model, similar to the results of previous studies done in German L1 speakers, and the specific areas of the brain reflecting language processing can be identified. However, if different language processing from that in German L1 speakers is seen, regular and irregular forms of plural noun endings in German L2 learners will be processed differently from those in German L1 speakers, and the areas of the brain that are activated will differ from those activated in German L1 speakers. As mentioned previously, this idea can be summarized by the following questions. 

The regular ending violation (*Minutes) will detect the LAN in morpho-syntactic processing at both levels. In terms of learning level, P600 detected sequentially after detecting LAN and, over some time, in high proficiency L2 learners compared to low proficiency L2 learners. No event-related potential components or LAN component will be detected in low proficiency L2 learners.

The irregular ending violation (*Auton) will detect N400 in morpho-syntactic processing that occurs upon violation of word/meaning; N400 will be detected at both levels. There will be no difference in the detection of ERPs by learning level.

In summary, low proficiency L2 learners process language according to the Full-Listing Model when dealing with irregular ending violations, but they will entirely fail to or insufficiently process ending violations according to the Full-Parsing Model compared to high proficiency L2 learners. In contrast, high proficiency L2 learners will show morpho-syntactic processing according to the Full-Parsing Model upon regular ending violations while following the Full-Listing Model when dealing with irregular ending violations. As a result, a difference in morpho-syntactic processing by learning level will be confirmed by detecting ERP components. It implies that high proficiency L2 learners use the Dual Mechanism Model compared to low proficiency L2 learners as the learning level becomes more difficult, and they perform morpho-syntactic processing similar to that in German L1 speakers.

## 2. Experimental Procedures

### 2.1. Participants

Twenty-six adult Korean native speakers (13 males and 13 females; age range 19–29 years; mean age = 23.82 years; standard deviation (SD) = 1.8 years) participated in the experiment. We tested a total of 26 L2 German speakers (with Korean as their L1). All participants had no history of neurological, hearing, speech, or psychiatric disorders. They had normal or corrected vision and were right-handed [27]. All participants were students at the university in Seoul and had no exposure to the German language before age 19. The participants were divided into two groups according to Zertifikat Deutsch (ZD) level and length of residence in countries where German was spoken: low (level range = below B2 in upper intermediate level), high (level range = above C3 in advanced level). Each participant was paid KRW 15,000 for their participation. Four participants were excluded from further analyses due to many artifacts in their EEG data (Table 1).

The study was approved by the Institutional Review Board (IRB) of Sungkyunkwan University, Seoul, South Korea (2014.09.01–2015.06.30). All participants signed a written informed consent form approved by the IRB (Approval No.: 2014-08-001).

### 2.2. Materials

German nouns were extracted from Schritte 1–6, published by Hueber. These words were compared by frequency in the Wortschatz database (Leipzig University) and consisted of 100 re-extracted words within classes 7–13 among 22 classes. We conducted a frequency survey of those 100 words in 120 students in the Department of German Language and Literature at a university located in Seoul. We classified words with >70% recognition as high familiarity and words with <50% recognition as low familiarity based on the survey. There were 28 critical nouns of 14 words. The critical nouns differed concerning noun plural type (regular (-(e)n) and irregular (-s)). These nouns were created by applying the violation paradigm using two types (grammatical vs. non-grammatical). As a result, it were selected in 4 conditions consisting of 14 words per condition. A total of 56 target sentences consisted of 6 words. The 56 target sentences consisted of four experimental conditions: (1) grammatical plural ending “-n” for female noun ending with “-e”, (2) ungrammatical plural ending “-n” for female nouns ending with “-e”, (3) grammatical plural nouns ending with “-s”, derived from the loan word as the critical nouns, and (4) ungrammatical plural nouns ending with “-s”, derived from the loan word as the critical nouns. The critical nouns were presented in the final sentence position. The 5th words were presented as numbers or quantitative adjectives to predict whether the critical noun was plural. The subjects of all sentences were presented as the names of German males and females. We also presented 84 filler sentences that were the same as the target sentences to hide the experiment’s intent (Table 2). This experiment used a total of 140 sentences, including 84 filler sentences and 56 target sentences. All stimuli sentences were randomly presented once each while preventing target sentences from being presented more than three consecutive times.

### 2.3. Procedures

The stimuli were presented using E-Prime 2.0 (Psychology Software Tools Inc., Pittsburgh, PA, USA) [28]. Sentences were presented one word at a time in yellow letters on a blue background to reduce eye strain, as identified in previous studies [29,30]. Each trial started with a fixation 『+』 in the center of the screen. The words were shown for 350 ms, and stimulus onset asynchrony between successive words in a sentence was 150 ms. The last critical nouns were presented for 800 ms. We adjusted the time duration of presentation of the critical nouns for L2 learners for Koreans, according to prior research by Weber and Lavric [31]. After presenting the last 6th, word, the stop mark 『•』 was presented for 2000 ms (Figure 1). During this time, the participants were asked to press “O” if the sentence was grammatically correct or press “X” if it was grammatically incorrect. We attached the labels “O” and “X” on the keyboard to avoid participant confusion. The program was set to automatically continue with the next sentence if the participant failed to respond within the given time. Each participant was tested in a single session of about 45 min, including placement of the electrodes. The procedures were done in a quiet laboratory at the Gangnam Severance Hospital of Yonsei University.

### 2.4. EEG Recording and Data Analysis

EEGs were recorded with a 64-channel HydroCel cap (Electrical Geodesic Inc., Eugene, OR, USA) with a 10/20 system. All electrodes were referenced to the left and right mastoids and re-referenced off-line to average references. In addition, the vertical electro-oculogram (VEOG) and the horizontal electro-oculogram (HEOG) were recorded to reject artifacts such as eye movements. EEG signals were amplified by the GES 400 Amplifier system. We applied a notch to block artifacts emerging behind 60 Hz. The EEG signals were digitized with a sampling rate of 500 Hz, and electrode impedances were kept below five kΩ. The EEG was recorded for about 15 min from the ground electrode installed on the common reference between Cz and Pz. It means standard EEG electrode names and positions. Cz is located in Midline Central. Pz is Midline Parietal.

In addition, we confirmed that the electrocardiogram (ECG) was one of the components that might directly influence the EEG in the pilot experiment. Heart rate variability (HRV) obtained from the ECG is the most massive biological signal in the body. It is measured as a regularly recorded signal with the EEG. It was presented with the ECG simultaneously at the time of critical noun presentation, and it may be considered an artifact and removed from ECG analysis. The ECG was recorded using a Physio Box (Electrical Geodesics Inc. [EGI], Eugene, OR, USA). The ECG was conducted by attaching ECG-specific patches on the 3 cm below the most prominent part of both sides of the clavicles. It was removed through QRS (QRS complex includes the Q wave, R wave, and S wave in ECG) and OBS (Quarterbacks of Heart Rate Variability) analysis before artifact removal during preprocessing.

### 2.5. Data Analysis

Data analysis was conducted using Netstation 5.0 software (Electrical Geodesics Inc. [EGI], Eugene, OR, USA). First, a bandpass filler was applied at 0.1–30 Hz. In addition, eye movements were extracted by artifact detection, and bad channel replacement was analyzed. We calculated the segment per condition. Then, the final data were calculated through an average of all participants. The average ERPs were calculated from −200 ms–1000 ms after the onset of critical nouns, including a 200 ms prestimulus baseline. The first 200 ms (−200 ms~0 ms) were not included in data analysis. With each condition containing a grammatical error, based on visual inspection of waveforms and topographic maps, we confirmed that the three different time windows and electrode sites captured language-related ERP components in the data. We focused the largest peak (or amplitude) after target word temporal regions where larger amplitudes were noted over all of the hemispheres. This means that when conducting experiments, the participant automatically skipped the words sequentially. We targeted the largest peak value when the target word was displayed.

We conducted a standardized LORETA (sLORETA, standardized low resolution brain electromagnetic tomography) analysis to scrutinize the time-rocked ERP components and brain regions. sLORETA is used to determine subcortical activity based on the EEG recorded from the surface [32]. It involves estimating the source of current density and calculation of an active current so it can be used to provide visualization of the brain in-depth.

Eighteen electrodes were organized into six regions of interest (ROIs) for ANOVA(Analysis of variance analysis): left anterior (LA: FT7, F3 FC3), midline anterior (MA: Fz, FCz, Cz), right anterior (RA: F4, FC4, FT8), left posterior (LP: TP7, P3, CP3), midline posterior (MP:Pz, CPz, Oz), and right posterior (꼐: P4, CP4, TP8), based on previous studies [24,25,26,27,33]. We performed repeated-measure ANOVA with components of learning level (low vs. high), regularity (regular vs. irregular), violation (grammatical vs. ungrammatical), hemisphere (left, middle, right), and anterior/posterior involvement. Repeated-measure ANOVA was used to compare three or more group means where the participants were the same in each group [34]. The significance level was set at *p* < 0.05. The data analysis was conducted with SPSS 21.0 (SPSS Inc., Chicago, IL, USA) [35].

## 3. Results

### 3.1. Behavioral Results

Both the low and high L2 groups achieved high accuracy rates in the grammatical judgment task for all conditions. Apart from the learning level, the highest correct answer rate was recorded under grammatical conditions, including plural nouns ending in “-s” in both groups. Under non-grammatical conditions of plurals ending in “-s”, the correct answer rate was approximately 4.0% higher in low L2 learners than high L2 learners. The result of the linguistic experiment is provided in Table 3.

### 3.2. Event-Related Potentials Results

#### 3.2.1. Irregular Ending Violation: N400

In cases of irregular ending violations (*Auton), we confirmed N400. It showed that the low proficiency L2 learners detected larger negativity about 482 ms after stimulus onset. According to its polarity, latency, and distribution, we termed this ERP component N400. In addition, high proficiency L2 learners appeared to have the greatest negativity at about 502 ms after stimulus onset, which was a bit slower compared to the low proficiency of L2 learners. Both groups were confirmed to have the CZ electrode site in the central parietal lobe. When detecting N400, the brain image obtained from sLORETA analysis showed the greatest activation in the central parietal lobe, about 482 ms after stimulus onset (Figure 2).

#### 3.2.2. Regular Ending Violation: LAN

For low proficiency L2 learners, we confirmed LAN in a regular ending violation (*Minutes).

There appeared a maximal negative effect at approximately 482 ms after stimulus onset. According to its polarity, latency, and distribution, we termed this ERP component LAN. In addition, the analysis showed that the greatest current density was present in the left frontal lobe at 482 ms, where the LAN was detected in low proficiency L2 learners. Similarly, high proficiency L2 learners had a LAN that was detected at the greatest negative waveform of approximately 555 ms after stimulus onset, which was a bit slower than that in the low proficiency L2 learners. When the greatest LAN is detected upon the violation, the brain image has detected both groups in the left frontal lobe. In comparison with grammatical conditions, we confirmed that different brain regions were activated at that time (Figure 3).

#### 3.2.3. Regular Ending Violation: P600

For high proficiency L2 learners, we confirmed P600 as a regular ending violation (*Minutes). The results show that P600 reached a maximum of about 686 ms after stimulus onset (*Minutes). Figure 4 shows the brain map of P600 detected at 686 ms after stimulus onset (*Minutes). Morphemic/syntactic violation for the critical noun was detected by LAN at 450 ms and sequentially at 600 ms. The brain image was obtained from sLORETA at the P600 detection time upon regular ending violation in the high proficiency learners. This image confirms that the greatest current density occurs at the domain behind the central parietal lobe in the high proficiency learners at P600 (Figure 4).

### 3.3. Statistical Results

#### 3.3.1. 300–450 ms

At 300 ms to 450 ms, the learning level difference under a non-grammatical condition (*Auton/*Minutes) was not significant. There was no difference by learning level, as both the low and high proficiency L2 learners showed the same response from LAN detection. However, a statistically significant result was found in the low proficiency L2 learners when comparing non-grammatical and grammatical conditions at the FC3 (F = 14.315, *p* = 0.007) and FT7 (F = 9.437, *p* = 0.032) electrodes. Therefore, it is presumed that the low proficiency L2 learners were determined to recognize morpho-syntactic violations under non-grammatical conditions, and the LAN or LAN-like EEG responses occurred at the FT7 as well as FC3 electrodes. Meanwhile, significance at the FC3 electrodes was not seen in high proficiency L2 learners, differing from low proficiency L2 learners; a statistically significant result was obtained from the FT7 electrode only (*p* = 0.011). In addition, the statistically significant result under violation conditions was obtained from the left anterio domain of the left frontal lobe from the FT7 electrode for six domains by ROI (F = 8.666, *p* = 0.018/F = 5.623, *p* = 0.046). In other words, the left frontal lobe was confirmed to be the location where morpho-syntactic processing of LAN occurs at 300–450 ms (Table 4 and Table 5).

#### 3.3.2. 450–600 ms

Statistically significant results by non-grammatical conditions and (*Auton/*Minutes) were confirmed at 450–600 ms (F = 5.758, *p* = 0.043). It is indicated that the EEG responses to two types of plural noun ending conditions were differently detected depending on the learning level. The statistical significance of the LA and MP domains, where LAN and N400 were detected, was confirmed in the low proficiency L2 (F = 5.758, *p* = 0.044/F = 5.417, *p* = 0.046) and the high proficiency L2 (F = 8.866, *p* = 0.012, F = 17.640, *p* = 0.031) learners. However, the statistical significance of N400 was not present in either the low or high L2 learners. The significance of the electrode location was confirmed at FT7 (F = 6.169, *p* = 0.033) and Pz (F = 5.623, *p* = 0.029). Based on these results, it is significant in terms of EEG and statistical analysis that both groups correctly recognized morpho-syntactic violations and, accordingly, LAN was detected. However, for N400, no statistical significance was confirmed even though N400 per learning level was detected in both groups. These results are summarized in Table 4 and Table 5.

#### 3.3.3. 600–800 ms

A significant difference under non-grammatical conditions (*Auton/*Minutes) was found by learning level, violation condition, and detection (F = 29.541, *p* = 0.001). It indicates that responses to violation conditions during the morpho-syntactic process of plural noun ending differ depending on the learning level, and the detection of P600 is directly associated with such morpho-syntactic processing. There was no ERP component detected in the low proficiency learners at this time point, and no statistically significant result was obtained. However, P600 has been detected in the high proficiency L2 learners. A statistically significant result was obtained from the LP (F = 5.770, *p* = 0.029) and the MP (F = 16.226, *p* = 0.002) brain domains. The result appeared in the high proficiency L2 learners only, suggesting that a definite difference in reprocessing develops during morpho-syntactic processing according to the learning level. In terms of the location of the electrode, significant results were confirmed at P7 (F = 7.781, *p* = 0.024), CP5 (F = 9.336, *p* = 0.015), and PZ (F = 3.151, *p* = 0.027). Electrodes were located a bit behind the central parietal lobe that was statistically significant in violation conditions. From such results, it can be determined that the domains in which the brain responds during morpho-syntactic processing occur beginning at 600 ms (Table 4 and Table 5).

## 4. Discussion

The present study was conducted on event-related potentials and aimed to examine whether the processing of regularly and irregularly inflected German noun plural endings are associated with distinguishable EEG signals. We investigated a sentence comprehension task to collect ERP components for different evidence employed by German L2 learners. In comparison to correct sentences, regular noun plural ending violations elicited left anterior negativity (LAN) in both levels.

ERP components confirmed that N400 during language processing in terms of vocabulary/meaning was detected regardless of learning level, and irregular violations occurred for plural nouns ending in “-n”. It indicates that both groups of German L2 learners recognized the irregular ending “-n” among plural noun endings as a meaning violation, not as a syntactic violation, suggesting that language processing supports the Full-Listing Model during morphemic processing. However, LAN and P600 detected from morphemic/syntactic processing imply differences by learning level. LAN detected from both types of German L2 learners confirms that regular ending violations occurred for plural nouns ending in “-s”. It shows that both types of German L2 learners recognize the regular ending “-s” among plural noun endings as syntactic violations, suggesting that language processing supports the Full-Parsing Model in morphemic processing. As P600 was additionally confirmed in the high proficiency L2 learners, such learners were found by P600 detection first to recognize syntactic violation and then to sequentially attempt reprocessing of syntactic violations. The results confirm that, as the learning level improves, different morphemic/syntactic processing characteristics appear. Generally, P600 is the morphemic/syntactic event-related potential component that was sequentially detected with LAN over some time and from the PZ electrode placed slightly behind the brain’s parietal lobe. Morphemic/syntactic violation for critical nouns was detected by LAN at 450 ms and then at 600 ms. It illustrates that reprocessing after morphemic/syntactic violation involved the anode placed slightly behind the central parietal lobe under non-grammatical conditions; the anode appeared greatest at the PZ electrode. Accordingly, L2 learners seem to use a Dual Mechanism regardless of learning level and more actively use it as their learning level increases. It can be considered that improvement in L2 learners’ learning level approaches cognitive processing similar to that used by German L1 speakers.

Our results showed that N400 and LAN among event-related potential components were detected at 450–600 ms in both types of German L2 learners. N400 was detected from the CZ electrode in the central parietal lobe, with no difference in terms of learning level, but the same N400 detection implies that plural noun irregular endings follow the language processing of the Full-Listing Model as the morphemic processing of vocabulary/meaning. LAN was detected from the FT7 electrode in the left frontal lobe, indicating that morphemic/syntactic violation of words included in the grammatically incorrect sentences was recognized, but there was no difference in learning level. However, irregular endings of plural nouns were found to follow language processes as per the Full-Parsing Model, which was confirmed by detecting the event-related potential factor, LAN, during morphemic/syntactic processing. In contrast, P600 has not been detected in the low proficiency L2 learners but was seen in the high proficiency L2 learners only at a specific time and domain (686 ms, PZ). It can be seen as a difference between the two types of L2 learners, as the high proficiency L2 learners recognized the words included in the grammatically incorrect sentence and accomplished reprocessing over some time. Therefore, P600 can be a supporting factor to show the difference in learning level. Our analysis results suggest that both types of Korean-speaking German L2 learners use the Dual Mechanism Model. Regular and irregular endings of plural nouns are similar in German L1 speakers, but the high proficiency L2 learners were found to use the Dual Mechanism Model, in contrast to the low proficiency L2 learners.

## 5. Conclusions

This paper presents second language processing regarding regular and irregular ending processing of German plural nouns in Korean German adult L2 learners. In this study, EEG signals measured in real-time were assessed, and whether or not the Dual Mechanism Model was used by German adult L2 learners was examined and compared with previous studies [16,18].

In conclusion, the directions for future studies, and possible utilization of the information contained herein are suggested, discussing some problems. Firstly, we considered a similar training period for adult Korean German L2 learners and high proficiency learners staying in a German-speaking country. However, since we strictly considered proficiency level and exposure duration from participants, it was hard to record the EEG data from subjects. If we gather more experimental groups, we will achieve definite differences between the low and high L2 learners in the case of Korea natives. Second, the experimental paradigm may apply to future research about German L1 speakers. We will also try to experiment with bilinguals between Korean and German. Lastly, the experimental paradigm method using EEG in this paper can be checked for similar results using brain imaging modalities such as position emission tomography (PET) with a high level of spatial resolution or functional magnetic resonance imaging (fMRI), magnetoencephalography (MEG), etc. Finally, if additional analyses of biosignals (ECG/HRV), classified as the subject, are removed, new results can be expected as they act as artifacts. In addition, we suggest our result to help to diverse language processing fields such as acquisition, education for children or foreigners, etc. Furthermore, we will conduct new analysis methods such as Statistical Parametric Mapping (SPM) [36], microstate [37], a time-frequency investigation, and source modeling based on oscillation [38] will be used to obtain new results in language processing.

## Figures and Tables

**Figure 1 brainsci-10-00866-f001:**
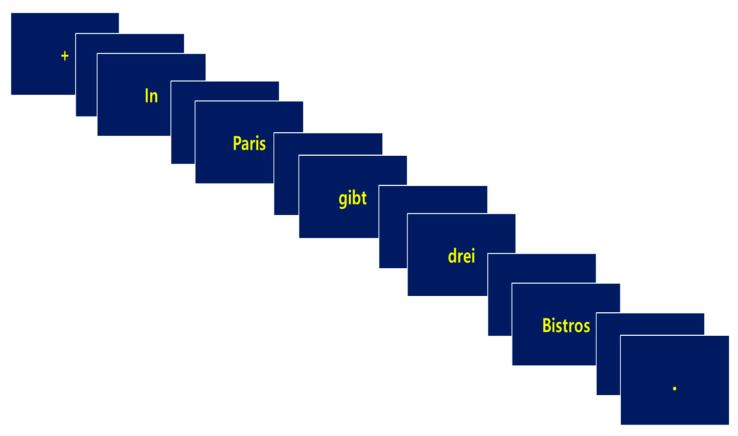
Schematic illustration of comprehension tasks.

**Figure 2 brainsci-10-00866-f002:**
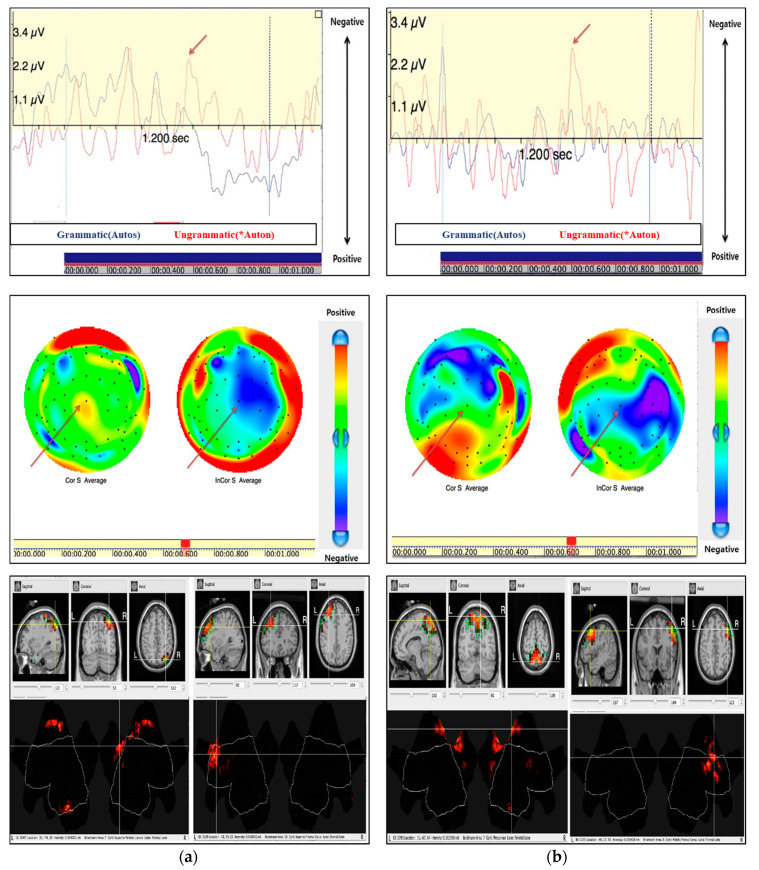
N400 (Cz) by irregular ending violation between low (**a**) second language (L2) and high (**b**) L2 learners.

**Figure 3 brainsci-10-00866-f003:**
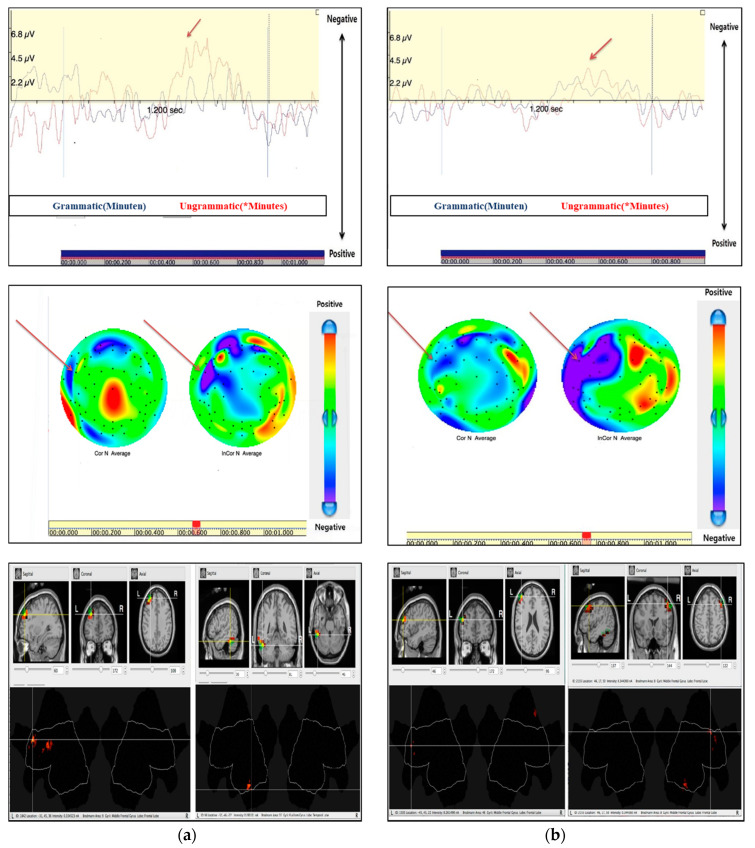
Left anterior negativity (LAN) (FT7) by regular ending violation between low (**a**) L2 and high (**b**) L2 learners.

**Figure 4 brainsci-10-00866-f004:**
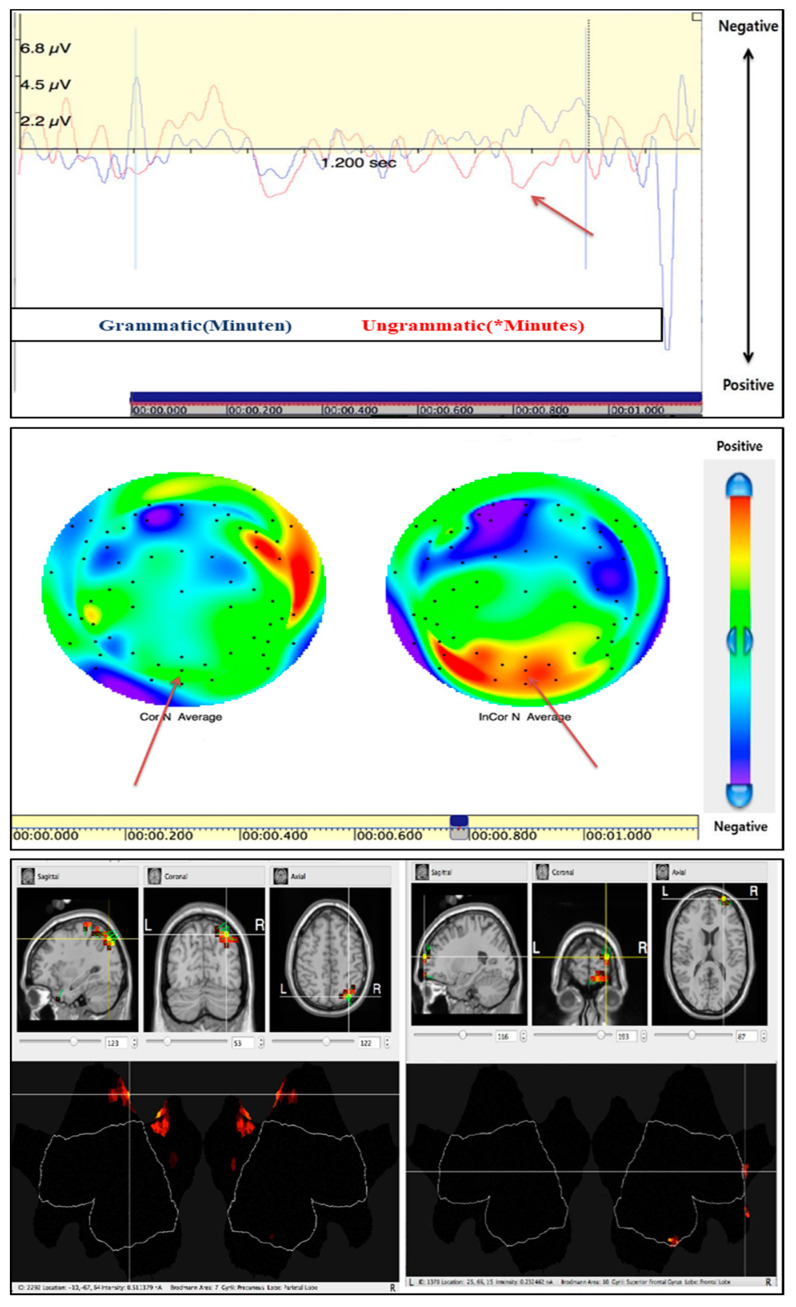
P600 (PZ) by regular ending violation in high L2 learners.

**Table 1 brainsci-10-00866-t001:** Demographic information of the study subjects.

	Low L2 Learners (*N* = 13)	High L2 Learners (*N* = 13)
Mean(SD)	23.5 (1.66)	23.9 (1.9)
Gender	M: 6/F: 7	M: 7/F: 6
Training period (year)	4.8 (2.1)	6.2 (2.3)
Exposure period (month)	1.1 (0.9)	10.2 (1.2)

( ) = Standard deviation.

**Table 2 brainsci-10-00866-t002:** Example sentences.

Plural Ending		Grammatical (Correct) Condition Sentence	*N*	Non-Grammatical (Incorrect) Condition Sentence	*N*
Regular	-s	Am Samstag kauft Jonas zwei **Autos**.	14	Am Samstag kauft Jonas zwei **Auton**.	14
		Jonas buys two cars on Saturday (English)			
Irregular	-n	Der Bus fährt alle zwangzig **Minuten.**	14	Der Bus fährt alle zwangzig **Minutes.**	14
		The bus comes to every twenty minutes.			
Filler		Im Zimmer gibt es alte Möbel			84
		There is old furniture in the room (English.)			
		Word: target word			140

**Table 3 brainsci-10-00866-t003:** Total number and percentage of correct answers for plural noun ending by learning level.

Plural Noun Ending		Low L2 Learner		High L2 Learner	
		Number of Correct Answers (*N*)	Percentage (%)	Number of Correct Answers (*N*)	Percentage (%)
**Grammatical condition**	-n (grammatical)	10.7 (2.5)	76.9 (16.2)	12.0 (2.4)	85.7 (15.3)
	-n (non-grammatical)	9.6 (2.1)	69.2 (15.6)	11.1 (2.3)	79.6 (16.7)
	-s (grammatical)	11.6 (1.2)	83.5 (8.3)	12.1 (1.5)	86.8 (10.8)
	-s (non-grammatical)	11.3 (1.7)	80.7 (11.3)	10.7 (2.2)	76.9 (14.5)
Total correct answers (*N*)		99.3 (14.2)	76.8 (9.5)	114.0 (11.9)	81.2 (11.4)

( ) = Standard deviation.

**Table 4 brainsci-10-00866-t004:** Event-related potential (ERP) results by learning level.

		Low L2 Learner		High L2 Learner	
		Grammatical	Non-Grammatical	Grammatical	Non-Grammatical
Plural noun ending “-n”					
LAN	Location of electrode	FT7			
	Time of detection	482 ms		554 ms	
	Difference in voltage	1.38 μV	−5.48 μV	−0.85 μV	−3.26 μV
P600	Location of electrode	Not detected		Pz	
	Time of detection			686 ms	
	Difference in voltage			−1.13 μV	3.63 μV
**Plural noun ending “-s”**					
N400	Location of electrode	Cz			
	Time of detection	482 ms		502 ms	
	Difference in voltage	1.35 μV	−2.39 μV	−0.42	−2.70 μV

**Table 5 brainsci-10-00866-t005:** ERP results using regions of interest (ROIs) by learning level.

		Low L2 Learner	High L2 Learner
300–450 ms	ROI (Region of interest)	Left anterior (FT7, F3, and FC3) (*p* = 0.018/*p* = 0.046)	
	Electrode	FT7 (0.007) FC3 (0.032)	FT7 (0.011)
	Learning level	No difference	
450–600 ms	ROI (Region of interest)	Left anterior (FT7, F3, and FC3) Midline posterior (Pz, CPz, and Oz)	
		*p* = 0.044/*p* = 0.046	*p* = 0.012/*p* = 0.031
	Electrode	FT7 (0.037)	FT7 (*p* = 0.033) Pz (*p* = 0.029)
	Learning level	difference (*p* = 0.043)	
600–800 ms	ROI (Region of interest)	Not detected	Left anterior (FT7, F3, and FC3) Midline posterior (Pz, CPz, and Oz)
			*p* = 0.029/*p* = 0.002
	Electrode		P7 (0.024) CP5 (0.015) Pz (*p* = 0.027)
	Learning level	Difference (*p* = 0.001)	

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
