# Peer review of "Morpheme Analysis Associated with German Noun Plural Endings among Second Language (L2) Learners Using Event-Related Potentials (ERPs)"

_brainsci, 2020, doi:10.3390/brainsci10110866_

Round 1

Reviewer 1 Report

The study investigates electrophysiological correlates inflectional morphology processing in L2 learners at two levels of proficiency. The purpose should be to experimentally approach the dual mechanism vs continuous model debate. Although the topic and research question are not extremely innovative, additional result might be of some interest. There are some substantial problems with this study though.

  • Introduction: "No event-related potential components or only LAN will be detected in low proficiency L2 learners" (lines 113-114). Why should only the LAN but not the P600 detected in low proficiency L2 learners?
  • EEG acquisition: arguably an online notch filter was used (line 189). Was a low-pass filter applied to. An anti-aliasing filter is necessary anyway.
  • ERPs: time window for analysis were arguably identified by visual inspection. This practice is nowadays highly discouraged. Time windows must be either a priori determined or other methodology must be adopted for the analysis (Jackknife, permutation test)
  • Was any statistical analysis on source data performed?
  • Statistical analysis: A repeated measure ANOVA was used. Sphericity must be tested and the results reported. If the assumption is violated, a correction must be applied. Moreover, in the Results p-values from what appears to be multiple post-hoc tests are reported. All these test must be corrected for multiple comparisons. F-values and effect sizes need to be reported.
  • The Conclusion section fails to actually conclude the paper. What are the final consideration the reader can retain from this study?

Unfortunately, in its current state the paper does not convey any reliable information because of both the lack of clarity and, most importantly, the methodological issues that make the results not interpretable.

As a minor remark, figures are crowded and hard to interpret, scales need to be made uniform and unit of measurement are missing.

Author Response

Dear reviewer.

We thank the editor for effective coordination and comments. This feedback is invaluable to help us take our research to the next level. Based on your comments, we revised the manuscript to explain the paper's concepts and findings in more detail. We also clarified the objective of the data analysis and reorganized the paper thoroughly. In particular, below are our detailed responses to the issues raised by the reviewers. You can find more information in the paper. Also, we have been thoroughly checked by professionals to accurately describe the grammar and expression used in this paper.

Reviewer 2 Report

This research study investigates the neural correlate of regular and irregular ending processing of German plural nouns using event-related potentials. ERP responses from 26 healthy control Korean-speaking German learners were examined. Participants completed a visual sentence comprehension task in which sentences were visually presented as single words and participants decided if sentences were grammatically correct, by a button press. Three N400, left anterior negativity (LAN), and P600 ERP components were found to be relevant to characterize the differences between low and high proficiency learners as well as regular and irregular ending nouns. N400 (central-parietal, 450-600ms) and LAN (pre-frontal, 450-600ms) were identified for irregular (meaning) violations regardless of learning level, whereas P600 (~700ms) was identified only in better learners, implied as a marker for learning level, in agreement with existing language theories such as dual mechanism model.

This work is well-written, details and explanations are clear, and findings are compelling,  I have provided some comments, mainly on the contents, methods, and results, hoping that they improve the quality of the work.

Abstract

  • Please add the full name of the LAN ERP component to this section.

Introduction

  • Line 74. I suggest using, “Electroencephalography (EEG) is an important functional neuroimaging tool for studying the temporal dynamics of human neural circuits” at the beginning of the paragraph, the first sentence.
  • Line 75. “ .. due to high accessibility ..” instead of “easy to access”.
  • Line 75. I suggest not mentioning (comparing against) other neuroimaging modalities, as it is beyond the focus of this work.

Section 2.4

  • Line 205, is it “from -200 to 1000 ms ..”?
  • Line 192. It is “Cz” not “CZ”. This applies elsewhere.
  • Line 1982. What is Com? Is it a common reference area? Please clarify.
  • Line 198. What is EGI? Please use the full name.
  • Please cite (Pascual-Marqui, 2002) for sLORETA.

Pascual-Marqui, Roberto Domingo. "Standardized low-resolution brain electromagnetic tomography (sLORETA): technical details." Methods Find Exp Clin Pharmacol 24.Suppl D (2002): 5-12.

Section 3.2.2

  • Line 259. Is it a typo, “Figure 16”?

Figs

  • 1. Please specify the timing (time intervals) on the figure and the caption.
  • Figs 2 and 3 are identical. Was that intentional? I feel that the figures are not well presented. This section requires revision.

Conclusions

  • Line 361. The term real-time could be misleading. Data were analyzed offline. I would not emphasize real-time.
  • Line 368. Please cite previous studies.
  • Line 377. Again, I would avoid the term “off-line”, unless your analyses were real-time (all data collections are real-time).

Final (general) note

  • Evoked responses mainly represent neural activity that is time- and phase-locked to the stimulus, responses that have significant jitter in their trial-to-trial latency, e.g., P600, are better detected (and localize) by averaging power changes in different frequency bands across trials (Pfurtscheller and Lopes da Silva, 1999). As future work, I suggest conducting a time-frequency investigation and source modeling based on oscillatory dynamics at a specific frequency range. For instance, beta-band, 18-24Hz desynchrony (compared against baseline) effects have been successfully used as a marker to localize responses relevant to language processing function, eg., see,

Youssofzadeh, V., Stout, J., Ustine, C., Gross, W.L., Conant, L.L., Humphries, C.J., Binder, J.R., Raghavan, M., 2020. Mapping language from MEG beta power modulations during auditory and visual naming. Neuroimage. doi:10.1016/j.neuroimage.2020.117090

Hope this helps.

Author Response

Dear reviewer.

We thank the editor for effective coordination and comments. This feedback is invaluable to help us take our research to the next level. Based on your comments, we revised the manuscript to explain the paper's concepts and findings in more detail. We also clarified the objective of the data analysis and reorganized the paper thoroughly. In particular, below are our detailed responses to the issues raised by the reviewers. You can find more information in the paper. Also, we have been thoroughly checked by professionals to describe the grammar and expression used in this paper accurately.

Reviewer 3 Report

The current study investigates the morpho-syntactic process of noun plural endings in Korean German learners, using event-related potentials. Authors hypothesized differences in language processing determined by learning level, and related differences in brain responses (such as the N400, the LAN and the P600) evoked by German plural nouns between high and low proficiency German learners. Results confirmed that, as compared to high proficiency learners, low proficiency German learners fail to process words by their components (i.e. stem and affix) but rather process words as a whole. In fact, low proficiency learners showed the N400 ERP component for irregular ending violation and the LAN component but not the P600 component for regular ending violation, supporting the idea that this type of word processing might follow the Full Listing Model. On the other side the P600 was detected for high proficiency learners, supporting the idea that their word processing might follow the Dual Mechanism Model.

The general idea behind the study is interesting, however I found the manuscript very hard to follow, because it requires significant editing; because methods, procedures and data analysis are unclear and barely reported; and because the results are not convincing and there are mistakes in the figures.  

Just to mention few concerns:

  • 13 participants were tested for each group, but then 4 participants were eliminated from the analysis (not clear from which group). This is a very low number of participants for an EEG study, especially considering that the authors attempted source-localization analysis using a 64 channels EEG system (more accurate localization of EEG data is usually achieved with high spatial sampling of the head surface electrodes). In fact, the EEG data appear to be extremely noisy from the figures that are reported in the manuscript.
  • Procedures and methods aren’t clear and it would be very helpful to list the sentences and the words used in the study in the supplemental material, with an English translation.
  • It is unclear how where the ROIs determined, as well as the temporal windows and the channel selected for the ERPs analysis (only through visual inspection? Of which group? Considering that there were differences between groups in amplitude, latency and topography).
  • There are no statistical results for the behavioral task.
  • The ERPs results are first reported by component and then by temporal windows, and I don’t understand this logic. The N400 should be merged with the 300-450 ms, the LAN with the 500-600 ms and the P600 with the 600-800 ms (again, not clear what type of analysis lead to the selection of these temporal windows).
  • How was the amplitude of the ERP components measured? Was the largest negativity compared or the average amplitude across the temporal window?
  • Not clear what the authors mean by “largest negativity”. Is it the most negative amplitude value in the full temporal window?
  • How was the latency of the ERP components calculated?
  • I’m skeptical about the filtering of the EEG data as from the figures showing the average voltage it is possible to notice some sort of oscillation. The noise might also be generated by differences across participants in each group (considering the low number of participants).
  • Figure 4 is the same of figure 3, and therefore the data from the P600 have not been showed.
  • In figure 3 the topographic plots seem to show the activity around 600 ms (P600 not LAN) while the voltage highlights differences around 500 ms.
  • Average amplitude values for each temporal window (and differences in the amplitude between incorrect and correct conditions) are not reported anywhere, even though the statistical comparison is made on these values (if I understood correctly).
  • Table 2 doesn’t report behavioral results, that is table 3.
  • The statistics (ANOVAs?) are poorly described.

Overall I believe that the manuscript requires a major editing and that at this stage it is hard to make an appropriate evaluation (especially considering the wrong figure and the missing information).

Author Response

Dear reviewer

We thank the editor for effective coordination and comments. This feedback is invaluable to help us take our research to the next level. Based on your comments, we revised the manuscript to explain the paper's concepts and findings in more detail. We also clarified the objective of the data analysis and reorganized the paper thoroughly. In particular, below are our detailed responses to the issues raised by the reviewers. You can find more information in the paper. Also, we have been thoroughly checked by professionals to describe the grammar and expression used in this paper accurately.
